# mmannot: How to improve small–RNA annotation?

**Matthias Zytnicki** *, **Christine Gaspin**

Unité de Mathématiques et Informatique Appliquées, Toulouse INRA, Castanet Tolosan, France

* matthias.zytnicki@inra.fr

## Abstract

High-throughput sequencing makes it possible to provide the genome-wide distribution of small non coding RNAs in a single experiment, and contributed greatly to the identification and understanding of these RNAs in the last decade. Small non coding RNAs gather a wide collection of classes, such as microRNAs, tRNA-derived fragments, small nucleolar RNAs and small nuclear RNAs, to name a few. As usual in RNA-seq studies, the sequencing step is followed by a feature quantification step: when a genome is available, the reads are aligned to the genome, their genomic positions are compared to the already available annotations, and the corresponding features are quantified. However, problem arises when many reads map at several positions and while different strategies exist to circumvent this problem, all of them are biased. In this article, we present a new strategy that compares all the reads that map at several positions, and their annotations when available. In many cases, all the hits co-localize with the same feature annotation (a duplicated miRNA or a duplicated gene, for instance). When different annotations exist for a given read, we propose to merge existing features and provide the counts for the merged features. This new strategy has been implemented in a tool, mmannot, freely available at https://github.com/mzytnicki/mmannot.

**Data Availability Statement:** Software is available at https://github.com/mzytnicki/mmannot.

**Funding:** The author(s) received no specific funding for this work.

## Introduction

Eukaryotic small RNAs (sRNAs) are defined as <200-bp long, usually untranslated, RNAs. They have been shown to participate in many aspects of cell life [1, 2].

They are generally classified according to their specific size range, biogenesis, and functional pathway. Among them, microRNAs (miRNAs) are certainly the most studied. Mature miRNAs are products of long primary transcripts, which fold into pre-miRNA stem-loop secondary structures that are recognized by the maturation machinery. Other sRNA classes can also be accurately identified, because they are part of a well-characterized structure: small nucleolar RNAs (snoRNAs), small nuclear RNAs (snRNAs or U-RNAs), and tRNA-derived small RNAs (tsRNAs).

On the contrary, other sRNAs have no known structure. Some classes are only defined by their positions relative to a protein coding gene, or a long non-coding RNA [3]. Some sRNAs are known to accumulate near TSS, near exon/intron junctions, in the promoter of the gene,

**Competing interests:** The authors have declared that no competing interests exist.

or in the downstream regions. Last, some classes have very few distinctive patterns (no clear secondary structure nor easily detectable primary transcript): small interfering RNAs (siR-NAs), piwi-associated RNAs (piRNAs) and repeat-associated siRNAs (rasiRNAs).

Although very diverse, these sRNAs can be captured in a single experiment, and several protocols have been devised to perform this task [4]. Briefly, the protocol includes RNA purification, size selection, possibly enrichment of terminal 5' phosphate molecules, and sequencing. Depending on the technology used, and the multiplexing, a few dozen of millions reads are produced.

It is then usual to assess the abundance of each sRNA class. To do so, the most straightforward way is to map the reads to a genome (usually with BWA [5] or bowtie [6]), and compare the mapped reads with an annotation file, which includes the positions of the known annotated sRNAs on the genome. This is the *annotation* step, which is executed for each read. The sRNA classes are then quantified by counting the number of reads co-localizing with the members of each class.

Although simple and widely used, this strategy does not work in several ambiguous cases, which are described in Fig 1.

1. A read maps at several loci: if two different regions of the genome are identical (usually after a genome duplication), a read may map equally well at different locations (Fig 1A).

2. Two different annotations overlap in the genome and a hit (i.e. a read mapping) overlaps both: In this case the hit may be attributed to either annotation (Fig 1B).

3. A hit co-localizes two different annotations, even though the annotations do not overlap themselves: The hit is usually at the frontier of the annotations (Fig 1C).

The first source of ambiguity is probably the most well-known and hard to tackle. Some sRNAs are known to co-localize with duplicated regions (such as piRNAs or siRNAs), or to be included into duplicated genes (miRNAs and tsRNAs). To circumvent these problems, several strategies were used:

- discard multi-mapping reads,

- use a random hit,

- weight each hit (if a read maps $n$ times, each hit counts for $1/n$).

However, every strategy presents obvious biases. The first strategy discards a usually sizeable proportion of the reads, the second one makes a random guess, and the last one may over-estimate the wrong classes, whereas it under-estimates the true class. Note that if a gene is duplicated, and a read maps uniquely to both of the duplicated genes, the two last strategies do work and provide the expected quantification.

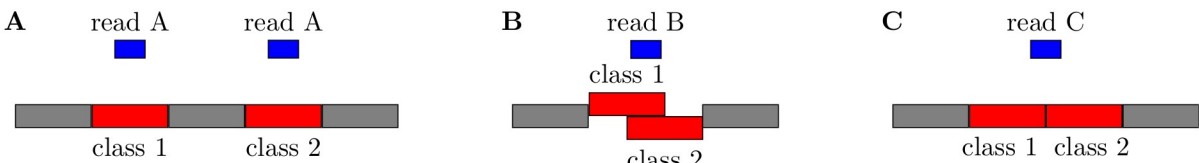

**Fig 1. Ambiguous cases of sRNA class quantification.** In each figure, the read is in blue, intergenic regions are grey, and known annotated sRNAs are red. *(A)* The read A maps at two different locations, and each mapping overlaps with a different sRNA annotation class. *(B)* Two sRNA annotation classes overlap in the annotation file. *(C)* The read overlaps two different sRNAs annotation classes that do not overlap each other.

There is, in principle, a fourth strategy, implemented by tools such as MMR [7]. MMR infers an optimal mapping location for a read, based on the expression of the putative *loci*. However, MMR, developed for long RNA sequencing, supposes that the distribution of reads is somewhat constant over a transcript. This is far from true in short RNA-Seq reads, especially for sRNAs silencing genes, where only a small region is targeted. Second, the real advantage of MMR is that it uses the coverage observed from non-ambiguous regions, to infer the read distribution over ambiguous regions. This strategy cannot simply work for miRNAs, where each read occupies the full length of the mature transcript.

A final strategy, implemented in ShortStack [8], adapts the previous idea to sRNA-Seq. Here, the estimation of the profile expression models the highly irregular expression profile, which is used to assign the multi-mapping reads.

Some papers, such as [9], use a radically different pipe-line. They first collect different nucleotidic databases (usually, one for each class), then map the reads to the databases. A read mapping to the database of a given class suggests that the read belongs to this class. However, this pipe-line does not solve *per se* the problem of the multi-mapping reads, since one read can map to several databases.

The second source of ambiguity, where several annotations overlap, may have two causes. First, there may be an inconsistency in the annotation. For instance, miRNAs are not known to co-localize with snoRNAs. However, we found that snoRNAs are sometimes detected as miRNAs, and so the two annotations may conflict in the same annotation data set. Second, the annotations may be on different levels. For instance, a miRNA may be localized inside an intron. If the user finds a read overlapping a miRNA which is an intron, the read will be probably attributed to the miRNA, and not to the intron.

The last source of ambiguity may arise when the annotations are not correctly defined.

Although these ambiguities do exist in the (long) RNA-Seq context, usual RNA-Seq quantification tools, such as featureCounts [10], cannot be readily used. First, these quantification tools count the number of reads per gene, and not per class. Second, regions producing small RNAs, such as introns and gene flanking regions, are not present in the annotation files, and are thus disregarded by these tools. Third, quantification tools usually expect the library to be stranded, or unstranded. Therefore, it is not (directly) possible to quantify reads which are antisense with respect to a gene, and the reads that can adopt any direction (for instance, in flanking regions), in a single run. Last, multi-mapping reads are usually discarded (this is the default behavior), or counted once per gene.

To solve these ambiguities, we propose a new strategy which is implemented into a sRNA class quantification tool, called mmannot. mmannot is freely available at https://github.com/mzytnicki/mmannot. In this work, we compared mmannot with the other five methods described here.

## Results

We compared our method with other possible approaches on several real-world, publicly available data sets, encompassing several organisms: *Arabidopsis thaliana* (see Fig 2), *Homo sapiens* (see Fig 3), *Danio rerio* (see Fig 4), and *Sus scrofa* (see Fig 5). We also benchmarked our tool on a synthetic data set that we generated (see Fig 6).

### Mapping only unique reads provides a highly distorted image of the sRNA class repertoire

Strikingly, the "unique" method provides very biased results in terms of representative percentage of the class. For the *A. thaliana* data sets (see Fig 2), the "unique" strategy annotates

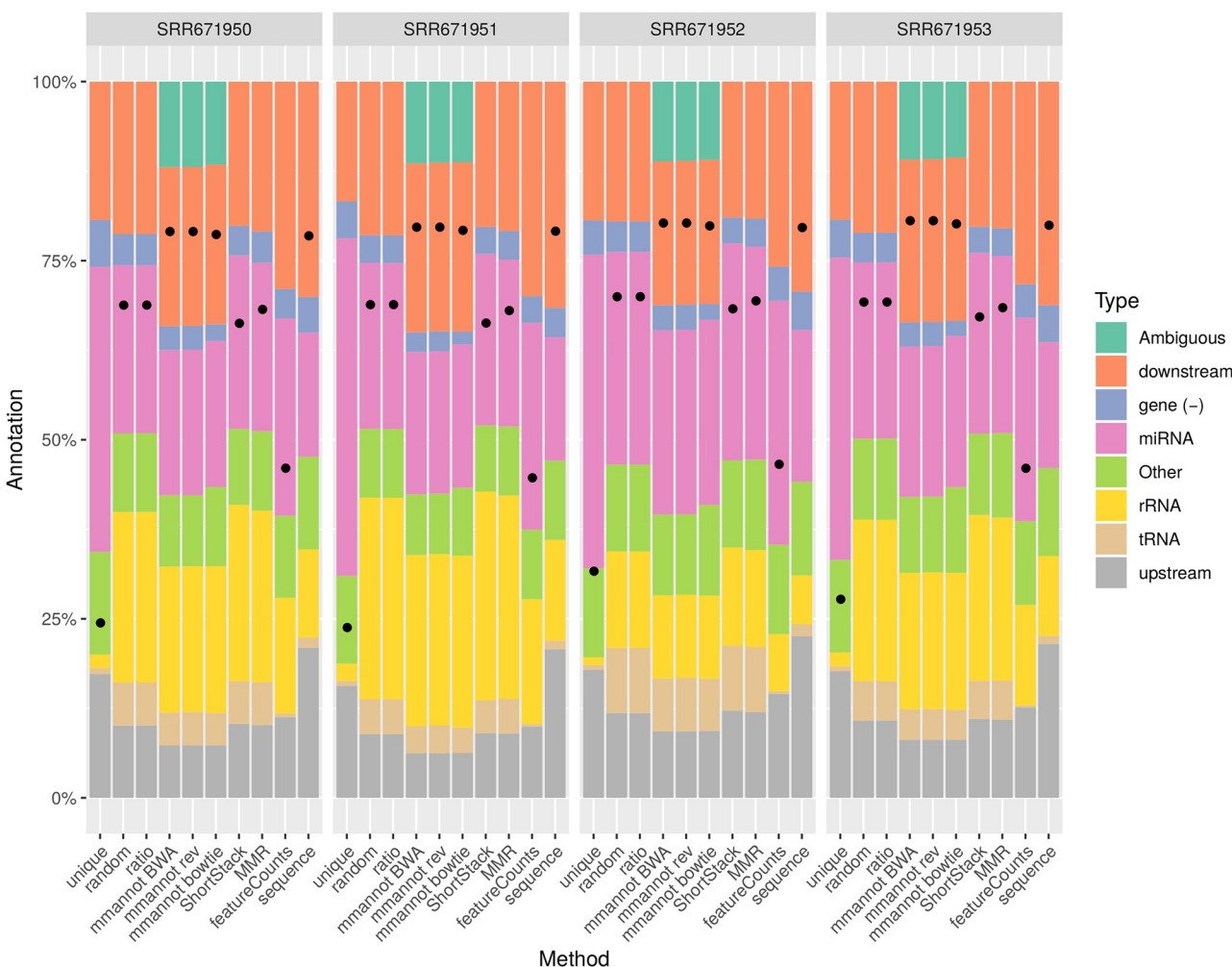

**Fig 2. Comparison of the different strategies on the *A. thaliana* data sets.** Each chart provides the result on one sequencing data set, using the previous strategies presented before. The "unique" strategy only considers uniquely mapping reads. The "random" strategy chooses arbitrarily a hit for each read. The "ratio" strategy counts 1/*n* per feature, if a read maps *n* times. Our method uses BWA for read mapping, and is named "mmannot BWA". In the "mmannot rev" lines, the order of the lines of the configuration files are reversed. The "mmannot bowtie" line gives the result of the "mmannot" strategy using bowtie instead of BWA for mapping reads. The "ShortStack", "MMR", and "featureCounts" columns gives the annotation results, based on their strategies. Last, we mapped the reads to sequences of sRNAs (instead of the genome), and provided the results in the "sequence" bars. Each column provides the percentage of reads, classified by annotation. The ambiguous category includes hits that overlap several annotations. sRNAs with low counts were grouped into the category "other". The dots in the bars provide the percentages of the reads that where annotated.

around 40% of reads as miRNAs, whereas the other strategies, when considering multi-mapping reads, annotate only around 20% of the reads in the miRNA class. Indeed, multi-mapping aware strategies can annotate the numerous reads produced by the highly repeated ribosomal RNA, and mmannot also annotates a high percentage of ambiguous reads. Moreover, we found that the "unique" method finds that between 126 and 137 miRNAs are expressed, whereas mmannot finds 149–159 expressed miRNAs. Whereas this difference is not great, one miRNA family, *miR157*, which contains three members exactly duplicated in the genome, produces 35k to 110k reads. Missing only one miRNA family (but several genomic loci) may have a dramatic influence on the results.

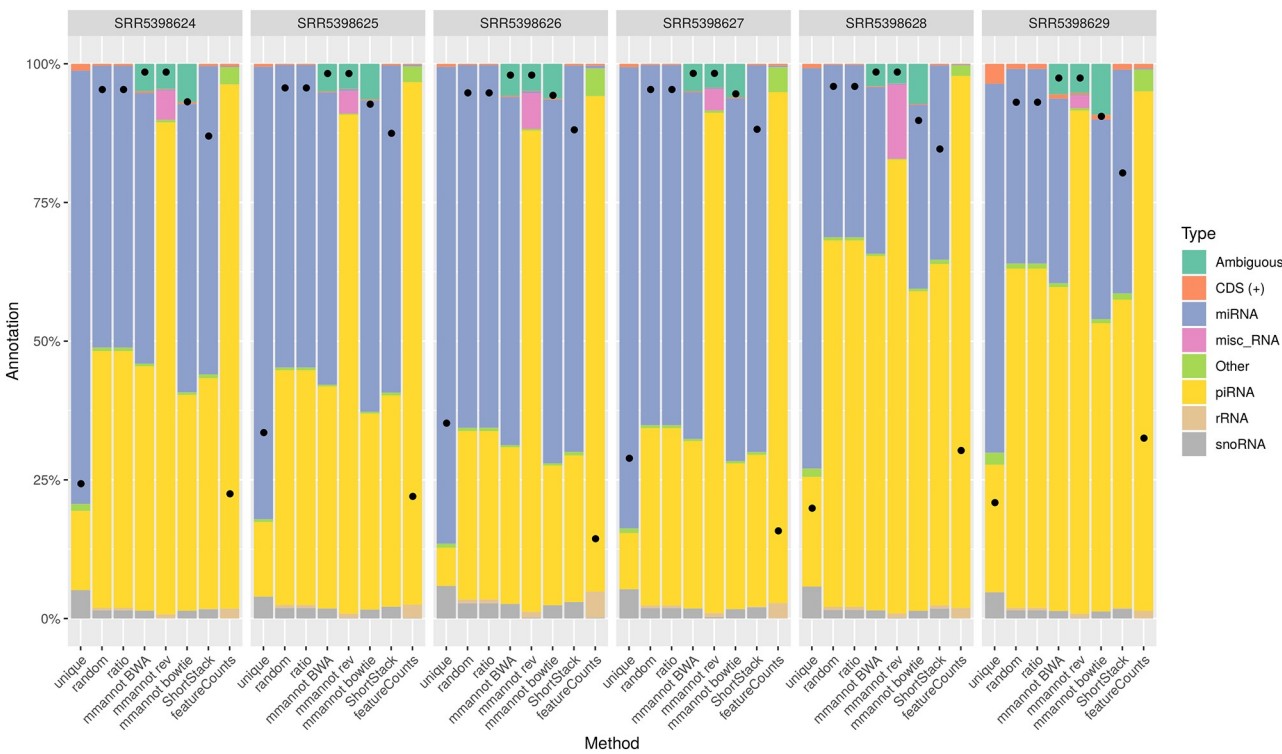

**Fig 3. Comparison of the different strategies on the *H. sapiens* data sets.** See Fig 2 for a legend of the plot.

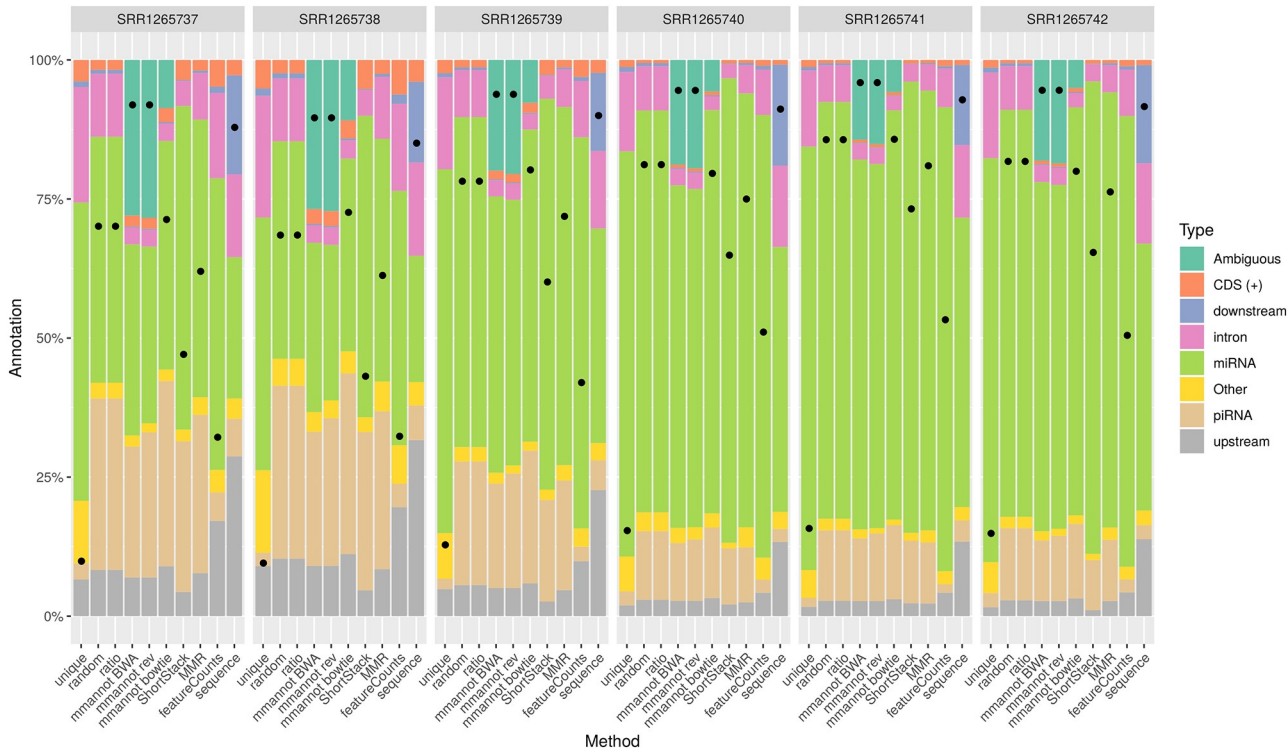

**Fig 4. Comparison of the different strategies on the *D. rerio* data sets.** See Fig 2 for a legend of the plot.

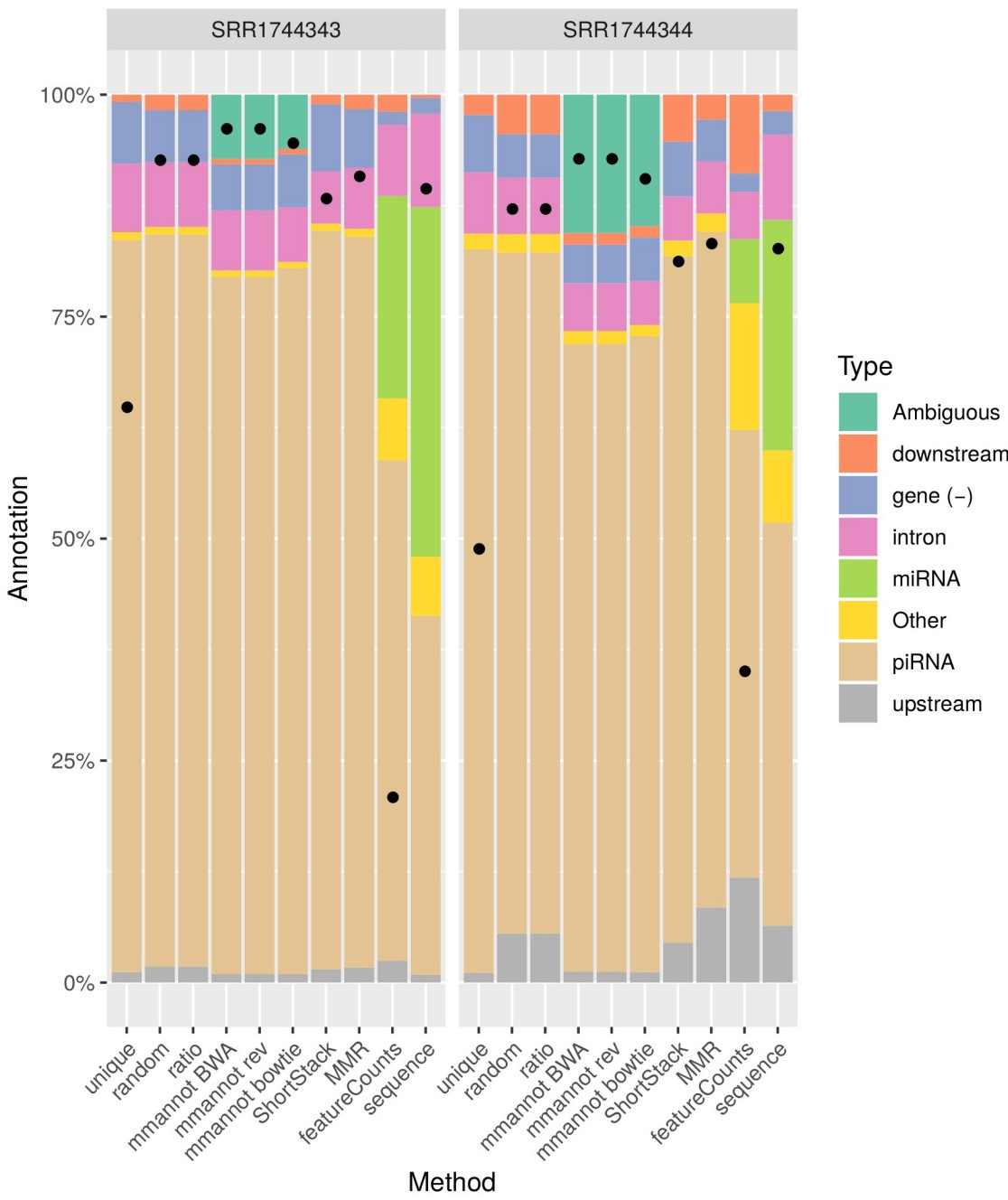

**Fig 5. Comparison of the different strategies on the *S. scrofa* data sets.** See Fig 2 for a legend of the plot.

Likewise, the proportion of miRNAs are over-estimated in the human and the zebrafish data sets.

As a result, using uniquely mapping reads only provides a highly distorted image of the complete sRNA class repertoire associated to a data set.

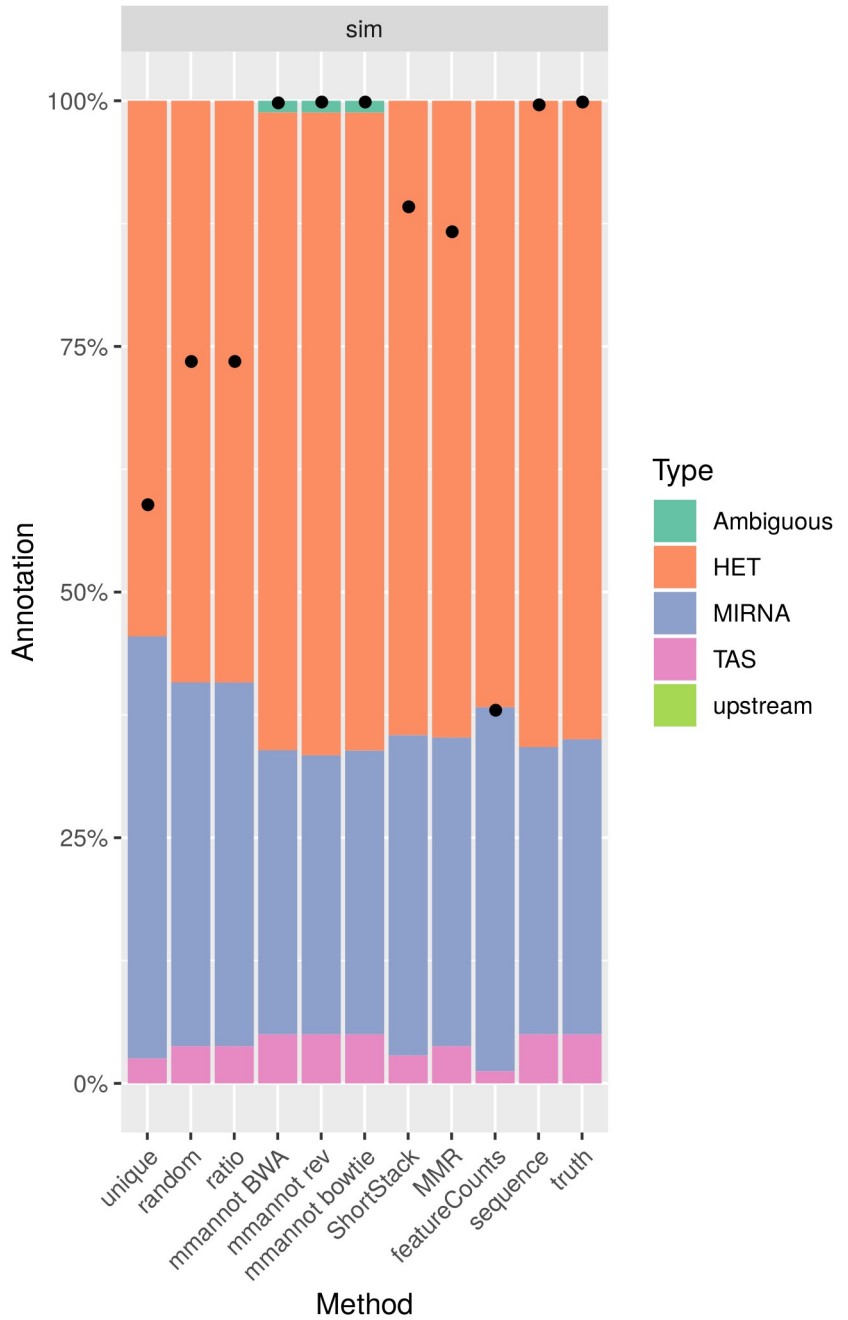

**Fig 6. Comparison of the different strategies on the simulated data sets.** The results of all but the last bars are explained in Fig 2. The "truth" bar gives the fraction of each sRNA class generated by our method.

## Multi-mapping aware strategies increase the number of annotated reads

For all data sets, using multi-mapping strategies increases considerably the percentage of annotated reads, showing that the repertoire of expressed regions is largely associated to repeated regions in all genomes. For the *A. thaliana* data set, the number of annotated reads is 0.8–1.8M using the "unique" strategy, whereas it is 1.4M–3.0M for featureCounts, 2.1M–4.4M

for MMR, 2.1M–4.3M for ShortStack, 2.2M–4.5M for the "random" and "ratio" strategies, and 2.5M–5.2M using mmannot. The "sequence" strategy gives similar results as mmannot.

All the other real-life benchmarks rank the tools similarly. The "unique" and "feature-Counts" use the least number of reads (since they behave similarly). "mmannot" and "sequence" use the greatest number of reads.

Some multi-mapping reads may have hits that do not co-localize with any annotation. These reads may be unannotated by the "random" strategy, and the associated weight in the "ratio" strategy is lost. These reads can also be misplaced by MMR and ShortStack. As a consequence, some of these reads are not quantified, resulting in less annotated reads. In the "mmannot" strategy, all the hits of a read are compared with an annotation, and used to annotate the read.

Fig 7 displays the actual annotation rate —the number of reads used for the annotation, divided by the number of reads sequenced— on all the datasets. Notice that the actual annotation rate is impacted by the sequencing quality, since low quality reads are removed during preliminary filtering (see Supporting Information for the quality filters used), and are often

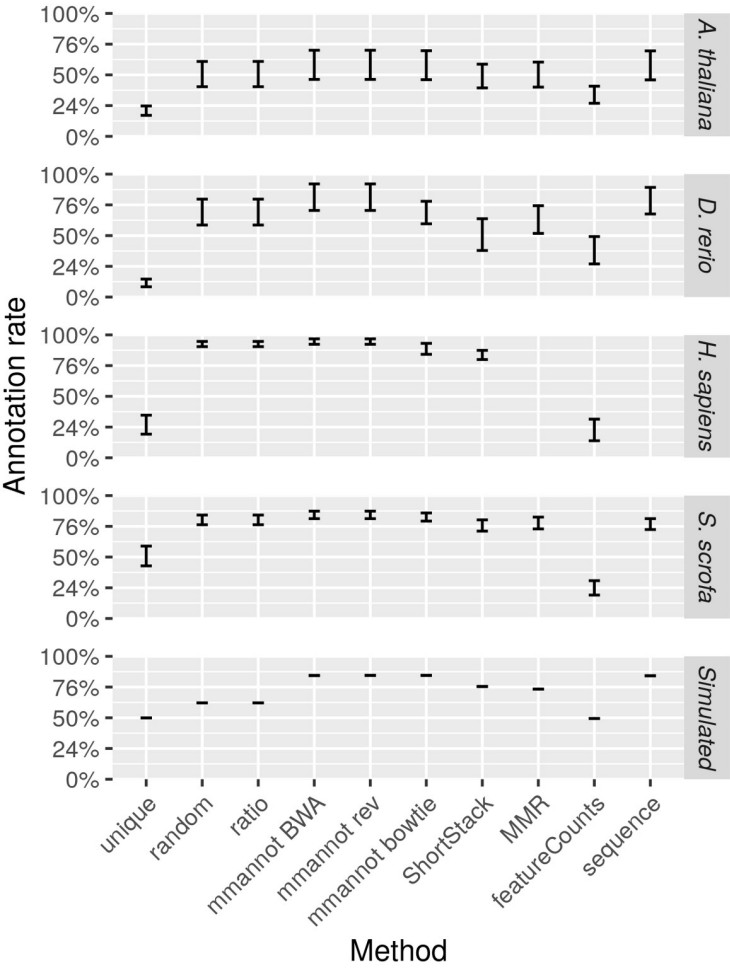

**Fig 7. Actual annotation rates of the methods.** The actual annotation rate is the number of reads used for the annotation, divided by the number of reads sequenced. The bars give the minimum and the maximum annotation rates for each method.

**Table 1. Statistical evidence that mmannot significantly more reads.** For each method, we tested whether our method uses significantly more reads with a one-sided Wilcoxon signed-rank test. Each column is a method to which we compared mmannot. The second row gives the p-value of the test. We did not compare with the "mmannot rev" method, because the number of reads used is exactly the same.

| method | unique | random | ratio | mmannot bowtie | ShortStack | MMR | featureCounts | sequence |
|---|---|---|---|---|---|---|---|---|
| p-value | $1.10^{-6}$ | $1.10^{-6}$ | $1.10^{-6}$ | $4.10^{-6}$ | $4.10^{-6}$ | $2.10^{-4}$ | $4.10^{-6}$ | $2.10^{-4}$ |

unable to map to the genome. The *H. sapiens* dataset provides the best annotation rate (almost up to 100%), because the quality of the reads is extremely good (we cannot exclude that low quality reads have been removed by the sequencing facility), and because the annotation is also exhaustive. Fig 7 confirms that mmannot gives the best annotation rates, together with the "sequence" method, while the "unique" and "featureCounts" methods provide worst results.

Finally, we wanted to confirm that mmannot uses significantly more reads than other methods. We performed a one-sided Wilcoxon signed-rank test between the number of reads used by mmannot on all datasets, and the number of reads used by the other methods. Results, shown in Table 1, clearly show that mmannot significantly uses more reads. Note, however, that the "MMR" and "sequence" methods seem to give slightly better results than other methods. This is however, only caused by the fact that these methods could not be used for the *H. sapiens* dataset. The test is thus less powerful in these cases.

Thus, mmannot improves considerably our knowledge of the structural and functional annotation of the gene repertoire associated to small RNA-seq data sets.

## mmannot controls the number of misplaced reads

The "random" strategy randomly places reads. As seen in Fig 8, this method yields a high number of false negatives. ShortStack and MMR can be seen as improvements of the previous method, as they try to place reads in the most likely locus. However, as seen on the same figure, these methods sometimes provide wrong results. About 10% of the reads are either placed in a locus that contains another annotation, or placed in a locus that contains no annotation. With mmannot, the right annotation is almost always given.

The "sequence" method maps many reads, but it also randomly attributes multi-mapping reads. Strikingly, the profile displayed by this method on the zebrafish and pig stands out, and are probably mis-annotations, although the synthetic data set does not detect many wrongly mapped reads.

## Both mapping tools yield similar results

Results on almost all datasets show that BWA maps slightly more reads. Moreover, BWA also provides much more hits per read than bowtie. This is a probably an advantage for BWA. However, the number of misplaced reads is close to zero in both cases (see Fig 8). Our results show that both tools give a similar picture of the sRNA repertoire.

The only exception is *D. rerio*, where BWA maps substantially more reads, and most of these reads are ambiguous. After inspection of these reads, we discovered that these reads match highly repeated regions: they match piRNA annotations, as well as gene flanking regions and introns. This confirms that BWA can find more hits per read than bowtie.

Note that choosing other parameters for the mapping tools could lead to dramatic changes in the results. Exploring the space of possible parameters is out of the scope of this article, even though we believe that the parameters we have chosen yield close to optimum results.

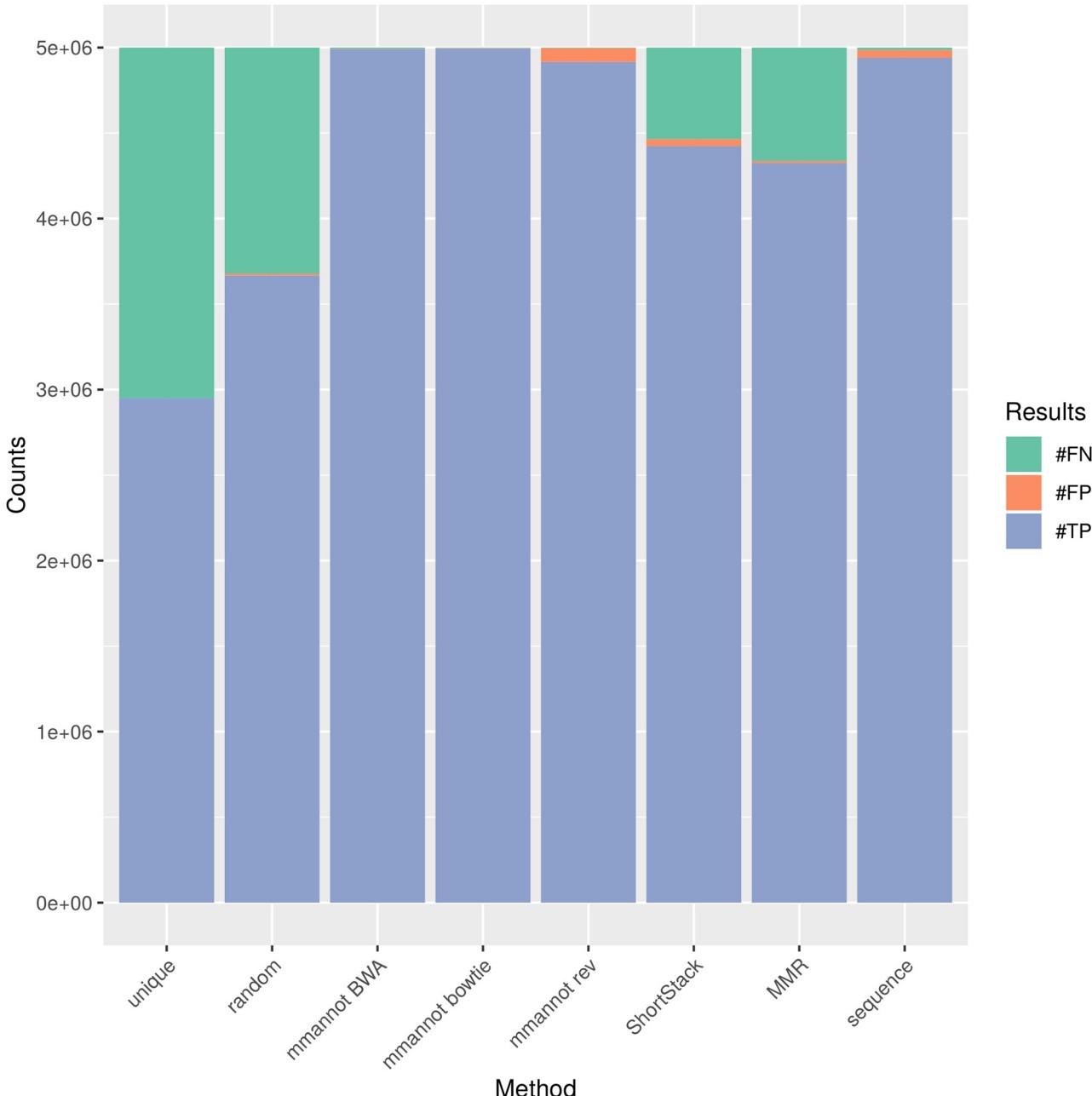

**Fig 8. Results on the simulated data set.** The number of true positives (#TP), false positives (#FP), and false negatives (#FN) are given for each method. We classified a multi-mapping read as true positive if at least one of the annotation given by mmannot is the correct one.

## Changing the order of the priorities does not change the quantification profile given by mmannot

The data sets show the quantification of the sRNA classes for two types of configuration files. The "mmannot BWA" uses a somewhat expected configuration file, where one read mapping both a miRNA and a piRNA would count for a miRNA. We reversed the ordering of the priorities in the "rev" columns. Strikingly, the results do not significantly differ.

The only exception is the human data set. Using the expected configuration file, the number of miRNAs is strictly greater than the number of piRNA. However, using the reverse configuration file, the number of piRNAs is much greater. To understand why, we compared the miRNA and the piRNA annotations. It turns out that 1178 out of the 3934 annotated miRNAs overlap a piRNA annotation. The most likely reason is that a great fraction of the piRNAs are probably mis-annotated, and using a meaningful configuration file is sometimes crucial for the analysis of small RNAs.

### mmannot predictions are more comprehensive

As seen on Fig 8, only mmannot and the "sequence" strategy give no false negatives. Of note, MMR and ShortStack quantifications are correct in general (see Fig 6). However, individual reads are sometimes attributed to the wrong class. This observation is a direct consequence of the wrongly guessed placement of the reads. This can be a problem when the user wants to inspect the localizations of a given multi-mapping read.

### All strategies leave a high number of reads unannotated

Whatever the multi-mapping strategy used, or the organism under study, all data sets show a high percentage of reads that are located in intergenic regions, located away from upstream and downstream regions, and for which no annotation is available (up more than 20% for *A. thaliana*). This lack of annotation is partially resolved with ambiguous annotations for *D. rerio* data sets (where about 25% of the reads are annotated as ambiguous).

### Analysis of the ambiguous reads of *A. thaliana* data set

We focused on the ambiguous reads of the *A. thaliana* data set in order to analyze their origin in that organism. Most of them come from downstream–upstream regions, which are probably intergenic duplicated regions. Then, the most frequent ambiguous annotations involve a transcribed region within the downstream, upstream, or intron regions. These elements might be produced by the unannotated regions to target genes, or belong to a non-functional genomic duplication. Interestingly, the most frequent ambiguous annotation involving two transcribed regions is miRNA–gene (−). In plants, a miRNA and its target may be 100% identical, and this poses a real problem to the annotation of the reads. We decided to study the pairs of miRNAs–genes that were involved in this class, with at least 100 reads supporting this association. We found well-known miRNAs and their targets: *miR156*/*miR157* with *SPL* [11], *miR163* with *PXMT1* [12], *miR171* with *ATHAM* [13], *miR400* with *PPR1* [14], *miR403* with *Ago2* [15] and *miR824* with *AGL* [16]. Note that these reads cannot be correctly annotated by any other method, and the expression of the miRNAs are thus under-estimated.

### Discussion

A high proportion of reads produced by sRNA-Seq maps to multiple loci, and this poses a major challenge to the annotation and quantification tasks. As a result, all the reads cannot be unambiguously annotated. To date, most of the tools do not report these ambiguous reads, because they remove these reads, or they weight the expression of the possible targets, using a model which can, or cannot be verified. We present here a new method, mmannot, which improves the sRNA annotation step, and improves the reliability of the small RNA feature quantification. The power of mmannot resides in a strategy that combines an effective use of multi-mapping reads and an on-the-fly annotation merging. Compared to the other strategies described in this paper, but also considering other strategies such as RSEM [17], the advantage

of mmannot over the other methods is sixfold. First, mmannot introduces no bias in the annotation, as it does not favor less duplicated classes. Second, it uses more reads than any other method to annotate the sRNA transcriptome. Third, it provides the proportion of reads that are actually ambiguous (something that the other methods cannot control). Fourth, mmannot has a better sensitivity and specificity than all other methods. Fifth, since it clearly pinpoints the annotations that are involved in these ambiguous reads, mmannot suggests candidates for possible regulatory RNAs, together with their targets. Sixth, it also identifies possible suspicious annotations, which are revealed by unexpected merged features. We hope that this method will contribute to the understanding of the repeated, dark matter of the genomes.

By nature, mmannot heavily rely on the accuracy of the available annotation. We saw, on the human data set, that the piRNA annotation might be flawed, and this dramatically alters the read annotation produced by our method. However, a simple, common-sense, ordering of the annotation types (favor "miRNA" when a read overlaps a miRNA annotation and a piRNA annotation) produces the expected results. Alternatively, discarding unexpected annotations (*e.g.* piRNAs in brain) might also be a sensible method.

In the future, we would like to find a way to optimally sort the annotation types. Most annotation methods, including mmannot, use simple rules for annotating the reads, mostly based on priorities given to annotation types. For instance, a read co-localizing with a miRNA and an intron is annotated as "miRNA". Whereas simple rules exist ("miRNA" > "introns"), other rules are arbitrary ("miRNA" is not comparable with "snoRNA"). In this article, we used an arbitrary ordering of the annotation types. We showed that mmannot gives similar results, even when the ordering is altered. However, finding a good set of rules would be desirable for this task.

## Materials and methods

### Solving different sources of ambiguity

To solve the first source of ambiguity, we provide a new method inspired by [18] which works on read annotation. Briefly, our method scans all the hits of a given read. If all the hits belong to the same annotation class and/or some of them are orphan of annotation, the read is annotated accordingly. Otherwise, the read is declared ambiguous, and all related annotations are provided. This method has several advantages:

- it is less biased,

- it makes use of all the reads,

- since ambiguous annotations are reported, related loci can be inspected,

- it makes use of all the hits.

The second source of ambiguity usually comes when different annotations classes overlap. In that case, our strategy consists in ordering the types of annotation. For example, users can decide that miRNAs have a higher rank than introns, and thus this ambiguity is solved. On the other hand, if miRNAs and another class such as snoRNAs have the same rank and a read overlaps both, the read will be declared ambiguous.

To solve the last source of ambiguity, where a read overlaps several annotations, the user can provide an overlap threshold, expressed in number of nucleotides or percentage of the size of the read. If the size of the overlap is lower than the threshold, the annotation is not considered. If several annotations can still be assigned to the read, the read is declared ambiguous, as previously.

**Fig 9. Detail of the mmannot process.** *(A)* Simple GTF file, which contains one coding gene and five miRNAs. *(B)* Excerpt of an mmannot configuration file. *(C)* Genomic localization of the annotation and the reads (for space reasons, the figure is not on scale). *(D)* Read annotation. *(E)* Quantification. *(F)* Feature quantification.

## Annotation strategy

mmannot requires three input files, and proceeds in three main steps (see Fig 9):

- First, mmannot parses a configuration file.

- Second, mmannot parses the annotation file (given by the user) and stores the annotation into memory.

- Third, mmannot reads the read file, and annotates each read.

Notice that some annotations are missing from a standard annotation file: introns and proximal regions are usually not mentioned, because they can be inferred from the provided annotations. To this end, mmannot automatically extracts introns of the features selected by the users, and adds them in the in-memory annotation data set (leaving the input file unchanged). Similarly, coding sequences (CDS), 5' and 3' untranslated regions (UTRs), down- and upstream regions of the annotations (the size of the down- and upstream regions are defined by the user) can be extracted. The user can also specify a strand orientation of the read with respect to the annotation (collinear or antisense). This way, intronic, sense, antisense, down- and upstream regions can also be quantified. Fig 9A presents a simple GTF file, which contains one coding gene and five miRNAs. The source field (the second field) is not provided, and the feature field gives the annotation type.

A configuration file (Fig 9B) is also required to select the annotations that should be quantified. The first section, `Introns`, provides the list of the features where the introns will be extracted. Here, a gene is automatically reconstructed from the GTF file, where the feature is `exon`. The next section, `Vicinity`, provides the list of the features where the up- and downstream regions will be extracted. The 5' and 3' UTRs of the gene also are automatically inferred. The last section, `Order`, provides the features that will be quantified. The colon (:) is the separator between the source field (*i.e.* the second columns of the GTF file) and the feature field (the third column of the GTF file). `.:CDS` thus means source. and feature `CDS`. As a result, if the line `.:CDS` is provided in the `Order` section, mmannot will count the number of reads that co-localize with all the annotations with feature `CDS` present in the annotation file. If a feature is followed by the plus (+) sign, like `.:CDS +`, mmannot will only count reads which are collinear with the annotation. In this context, a read in the plus strand overlapping with a CDS in the minus strand would not be used for the quantification. If a feature is followed by the minus (−) sign, mmannot will only count reads which are antisense with respect to the annotation. The annotation are ordered by priority, so that `CDS +` is before `intron` in the example given in Fig 9B. However, since `CDS +` and `5'UTR` are on the same line, they have the same priority.

Read annotation proceeds in two steps. The first step aims at finding the matching annotations of a given hit (Fig 9C). If a hit matches several different annotations, the annotation with highest priority is kept (e.g. read C in the example). If several annotations have the same priority, then all of them are kept and the hit is already ambiguous (read E in the example). In the example, the annotations related to the gene are provided in shades of green and black. The miRNAs are in red, and reads, in yellow, are labelled A to G. Note that reads D, F, and G appear twice, because they can be mapped at two different locations. The reads are supposed to be stranded, and the arrow indicates the strand.

The second step resolves the ambiguities and annotates each read. The general method is:

- If a read maps uniquely, with no ambiguity, the count of the corresponding annotation is incremented.

- If a read maps at different locations, but all the hits match the same annotation, the read is declared *rescued*, and the corresponding annotation count is also incremented (read D). Likewise, if a read maps only one annotation and intergenic regions, then the read belongs to this annotation, and the read is rescued.

- If the read or a hit overlaps several annotations, the annotations with highest priority are kept. However, if there is only one annotation with highest priority, the read or the hit is not ambiguous.

- Otherwise, there is an ambiguity. A new annotation type, called a *merged annotation*, is created. It is the concatenation of the matching annotations (as suggested in [18]), and its count is incremented.

The annotation table (Fig 9D) can be provided by mmannot.

In the example, the read A only maps one region, which only overlaps the upstream region of the gene. It can thus be unambiguously attributed to this feature. The read B uniquely overlaps the CDS of the gene. However, the read is in the − strand, and the gene is on the + strand. The configuration file specifies that only collinear reads should be attributed to the CDS, and thus this case cannot apply. However, the line `gene −` of the configuration file qualifies and the read is attributed to this feature. The read C uniquely maps to a miRNA and an intron. Since the miRNA has a higher priority, the read is attributed to this feature. The read D maps

at two different locations, but both these locations overlap a miRNA. The read can still be attributed to a miRNA, and mmannot mentions that this read has been *rescued*. Read E overlaps a CDS and a miRNA, and these annotations have the same priority. The ambiguity cannot be resolved, and we attribute E to `CDS (+)-miRNA`. Read F maps at two different locations. The first location overlaps the 3'UTR, whereas the second location overlaps a miRNA. Again, the ambiguity cannot be resolved, and we attribute F to `3'UTR (+)-miRNA`. Last, G overlaps the 5'UTR and an intergenic region. In this case, we assign G to 5'UTR and we declare the read rescued.

After having scanned all the reads, the quantification table is produced (Fig 9E).

The feature quantification table (Fig 9F) can also be provided by mmannot. It counts the number of reads per (possibly merged) feature. Here, the user can pinpoint that there may be some interaction between miRNA E and the 3'UTR, and a possible inconsistency between the annotations of the CDS and miRNA B.

## Details of the implementation

The algorithm used by mmannot is the following. mmannot scans the annotation file, extracts annotation intervals, sorts them, and stores them into non-overlapping bins. mmannot then reads the mapped read file, extracts the possibly overlapping bins, and compares the related annotations with the read. The matching annotations of a given read are kept into memory. When the last hit is scanned, the count of the (possibly merged) annotation is increased, and the read information is discarded from memory. In practice, the search is approximately linear with respect to the number of reads. mmannot can run on a standard computer, and each SAM/BAM file can be scanned by a dedicated thread.

mmannot either supposes that the `NH` tag of the input SAM/BAM mapped read file provides the number of hits per read, or that the alternative hits are given in the `XA` tag. Unfortunately, some widely used mapping tools such as bowtie [6] use one line per hit, and do not set the `NH` tag. We provide a companion tool that reads the SAM output file of bowtie and adds the correct `NH` tag. mmannot may produce three different output files.

The first file is the count table, which includes merged annotations. The second (optional) file provides the quantification of all the annotations (Fig 9F). This file provides detailed information of merged features, and makes it possible to investigate possible interactions between the features, or to correct the annotation. The third (optional) file gives the matching annotations of every read (Fig 9D). This file makes it possible to manually inspect all the reads that support a merged feature.

## Data used

We used published data sets, covering several eukaryotic organisms, where the related articles provided a quantification of the expression of the sRNA classes. Table 2 summarizes the data used.

The *H. sapiens* data set originally contained 16 samples. We display here only the five first samples, for the sake of brevety. The other sample produce similar results.

Note that at most 10% of the reads are unmapped, and the average percentage of unmapped is around 3%. Although not analyzing unmapped read may bias the results of the study, we believe that the impact is greatly limited.

In this benchmark, we recorded the time spent by mmannot with the output statistics files option enabled. These files take substantial time to be computed, and mmannot is much faster with default options.

**Table 2. Data used.** For each experiment, the reads were annotated using mmannot and available annotation files. The meaning of each row follows. 1: Organism. 2: SRA id. 3: Related publication. 4: Tissue from which the RNA was extracted. 5: Number of sRNA-Seq data sets. 6: Number of reads per sample. 7: Percentage of reads which passed the quality threshold. 8: Number of mapped reads (given by BWA) in each data set. 9: Percentage of unmapped reads, *i.e.* the proportion of reads that cannot be placed on the genome. 10: Number of hits, *i.e.* the number of possible positions for the reads (given by BWA). 11: Number of features in the annotation, extracted from the annotation file. 12: Time spent by mmannot for each data set, in minutes.

| Organism | *A. thaliana* | *D. rerio* | *H. sapiens* | *S. scrofa* | Simulated |
|---|---|---|---|---|---|
| SR | SRR671950 | SRR1265741 | SRR5398624 | SRR1744343 | — |
| Ref. | [19] | [20] | [21] | [22] | — |
| tissue | roots | brain | amygdala | testis | — |
| # data sets | 4 | 6 | 6 | 2 | 1 |
| # reads | 4–8M | 3–15M | 12–17M | 6–7M | 5M |
| % quality reads | 72–89% | 76–99% | 95–98% | 92–94% | 100% |
| # mapped reads | 3–6M | 2–13M | 10–17M | 6–7M | 5M |
| % unmapped reads | 1% | 2–3% | 5–10% | 3% | 0% |
| # hits | 12–26M | 102–530M | 39–111M | 19–28M | 40M |
| # features | 512,908 | 4,372,617 | 13,040,803 | 2,136,213 | 5,770 |
| time | 1–4 | 11–51 | 2–66 | 12–19 | 6 |

The most recent annotation, from Ensembl v69 [23] was downloaded, except for *A. thaliana* (we used TAIR 10 [24]). We extracted all the sRNAs from the annotation, as well as CDS, UTRs, introns, and proximal regions of the coding genes (± 3kb).

For *D. rerio*, *H. sapiens*, and *S. scrofa*, we downloaded a piRNA annotation from piRBase [25], and we added this annotation to the existing annotations (which usually do not include piRNAs).

## Benchmarking

We computed an exhaustive benchmark of our method, which assesses its performance and its robustness. Our aim was to show how each annotation strategy impacts the results of quantification in each class. We benchmarked several complex organisms, from the animal and plant reign (see Fig 2, 4, 3 and 5). We also added a synthetic data set, where the truth in known (Fig 6).

**Evaluated features of mmannot.** **Comparison on several organisms**. We compared all the methods on several organisms: *A. thaliana*, *D. rerio*, *H. sapiens*, and *S. scrofa*. These model organisms are well annotated, include a wide diversity of small RNA classes, and contain a high proportion of duplicated regions. They constitute a complex testbed for the annotation problem.

**Comparison on a simulated data set**. We generated a synthetic data set using the method developed to evaluate ShortStack [8]. This method provides both the reads, and the annotation. However, the annotation only consists of three types of sRNAs, and very few reads were ambiguous, contrary to our observations on the real data sets. Fig 8 provides accuracies and specificities of the methods.

**Comparison with other approaches**. We first compared our results with three strategies mentioned previously also implemented in mmannot: the "unique", the "random", and the "ratio" strategies.

We also mapped our reads using ShortStack [8], which provides an informed placement of the reads. ShortStack has been developed for sRNAs.

We used MMR [7], which also provides an informed placement of the reads. Contrary to ShortStack, MMR was developed for messenger RNA-Seq. MMR was used on the reads

mapped with bowtie, because it does not interpret correctly the alternative read hits as given by BWA. MMR requires a substantial amount of RAM. It exited with memory error on the human data set, after requiring more than 32GB of RAM.

We also compared our tool with the widely used featureCounts [10], with the caveats we mentioned in the Introduction. First, we extracted the small RNA classes from the annotation file, added missing features (introns and flanking regions), and generated another annotation file in SAF format (read by featureCounts). An example script (used for the *A. thaliana* data set) is given as Supporting Information. Second, we used featureCounts as usual. Of note, featureCounts either discards multi-mapping reads, or counts one hit per mapping gene. We chose the first option.

We also implemented a last strategy. We extracted the sequences from the previously generated annotation file, and the genome sequence. We thus built a nucleotidic database of the sequences of the regions of interest. We then mapped the reads to this database. However, we noticed an unwanted behavior: some reads, which mapped with no error on the genome (this information was given by the standard genome mapping strategy we used for mmannot), mapped with one or more errors on the database. As a result, this strategy classifies them as belonging to one of the known class, whereas they most likely come from some unknown locus. In order to avoid this bias, we also included the whole genome in the database. If a read maps to the genome and a known sRNA class (with the same number of errors), the reads will be attributed to the sRNA. If a read maps to the genome with $n$ errors, and a known sRNA class with $m > n$ errors, the read will be declared unannotated. However, this strategy does not solve the multi-mapping problem, and a read can map to several sRNA classes. In this case, we attributed the multi-mapping read to the first class given by the mapping tool (here, BWA). The code used to generate the database, and map the reads is given in Supporting Information. Of note, this strategy requires substantially much more memory, since each read may map to the genome, and its sRNA sequence, on both strands. As a result, we could not use it on our computer on the human data set, because it required more that 32GB or RAM.

**mmannot robustness to the ordering of the features**. The configuration file sorts all the features that will be annotated and quantified (e.g. miRNAs, piRNAs, etc.). By essence, every priority scheme between the features is arbitrary. We prepared new configuration files, where the order has been reversed: all the features that were top priority are bottom priority. However, untranscribed regions (introns, downstream, upstream, intergenic) still have the lowest priorities.

**mmannot robustness to the mapping tool**. The community mainly uses two mapping tools, BWA [5] and bowtie [6]. As suggested by [26], we selected BWA as our primary mapping tool, but we still compared the results with bowtie.

## Data preparation

Data were imported from SRA with fastq-dump [27]. Adapters were trimmed with fastx clipper [28]. Only sequences with sizes greater than 14bp were conserved. Low complexity reads were removed by using dust [29]. Remaining reads were mapped with BWA [6]. We also mapped the reads with bowtie [6]. The NH tag was added to the SAM file provided by bowtie. Files were then sorted and converted to BAM with samtools [30]. Quantification of the expression of the sRNA classes was realized with mmannot using "unique", "random" and "ratio" strategies. Our method is named "mmannot". The mapped reads were also filtered with MMR, which gives uniquely mapping reads, then quantification was done with mmannot. We also compared mmannot with ShortStack, which also assigns reads to a unique locus, and with featureCounts, which discards multi-mapping reads. Reads were also mapped to the sequence of

sRNAs using BWA (the "sequence" strategy). The simulated data set was generated by the script `sim_sRNA_library.py` [8], which was trained on our first *A. thaliana* sRNA-Seq data set and miRBase v22.1 [31]. We added coding genes, pseudogenes, and transposable element annotations (given by TAIR10) to the `MIRNA`, `TAS` and `HET` annotation (generated by the script).

All the experiments were performed on a machine with 6 Intel Xeon CPU E5-1650 v4 cores at 3.60GHz and 32GB RAM running Ubuntu 19.04.

All the commands used in this analysis are given as Supporting Information.

## Supporting information

**S1 File. Version of the tools, and scripts used in the benchmark.**
(PDF)

**S2 File. Raw expression quantification given by each method.**
(GZ)

## Author Contributions

**Conceptualization:** Matthias Zytnicki, Christine Gaspin.

**Software:** Matthias Zytnicki.

**Validation:** Matthias Zytnicki.

**Writing – original draft:** Matthias Zytnicki.

**Writing – review & editing:** Matthias Zytnicki, Christine Gaspin.

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
