## [Decision Letter · Decision Letter 0]

31 Jul 2019

PONE-D-19-16932

mmAnnot: How to improve small–RNA annotation?

PLOS ONE

Dear Dr Zytnicki,

Thank you for submitting your manuscript to PLOS ONE. After careful consideration, we feel that it has merit but does not fully meet PLOS ONE’s publication criteria as it currently stands. Therefore, we invite you to submit a revised version of the manuscript that addresses the points raised during the review process.

Reviewers have evaluated your manuscript but major changes are necessary to be acceptable for publication. Reviewers have provided detailed comments and suggestions. I invite you to revise the manuscript accordingly, in particular a main modification required is an extension of small RNA classes considered in the tool. Specifically, piRNA and tRNA-derived fragments should be included among the available categories. In addition the comparison with other available tools is not satisfactory, more tools should be considered.

We would appreciate receiving your revised manuscript by Sep 14 2019 11:59PM. To enhance the reproducibility of your results, we recommend that if applicable you deposit your laboratory protocols in protocols.io, where a protocol can be assigned its own identifier (DOI) such that it can be cited independently in the future. For instructions see: http://journals.plos.org/plosone/s/submission-guidelines#loc-laboratory-protocols

We look forward to receiving your revised manuscript.

Kind regards,

Francesca Rizzo, PhD

Academic Editor

PLOS ONE

Journal Requirements:

Please provide an amended Funding Statement that declares *all* the funding or sources of support received during this specific study (whether external or internal to your organization) as detailed online in our guide for authors at http://journals.plos.org/plosone/s/submit-now.  Please state what role the funders took in the study.  If any authors received a salary from any of your funders, please state which authors and which funder. If the funders had no role, please state: "The funders had no role in study design, data collection and analysis, decision to publish, or preparation of the manuscript."

Reviewers' comments:

Reviewer's Responses to Questions

**Comments to the Author**

1. Is the manuscript technically sound, and do the data support the conclusions?

Reviewer #1: Partly

Reviewer #2: Partly

2. Has the statistical analysis been performed appropriately and rigorously? 

Reviewer #1: Yes

Reviewer #2: Yes

3. Have the authors made all data underlying the findings in their manuscript fully available?

Reviewer #1: No

Reviewer #2: Yes

4. Is the manuscript presented in an intelligible fashion and written in standard English?

Reviewer #1: Yes

Reviewer #2: Yes

5. Review Comments to the Author

Reviewer #1: In this manuscript the authors proposed a tool to annotate small RNA based on user given annotation file. Their strategy is outputting all the annotation information to those small RNAs that mapped to genome. Although this strategy may work in several ambiguous cases, there are several concerns before acceptance this paper.

1. The intrinsic flaw of the tool leads to the result that it can only annotate RNAs that mapped to genome. However, 10% or more RNAs cannot map to genome due to post-transcriptional splicing/modification or other mechanism, which resulting in reduced accuracy of the tool. In this case, all the analysis in this work are biased.

2. The authors mentioned piRNAs and tRNA-derived fragments in the manuscript, but didn't show them in the results, are they classified in the ambiguous category or others?

3. Although the authors emphasized the advantage of their annotation strategy, the accuracy is based on the reference file the user choose. The annotation category of Ambiguous could be very high when adding piRNA database into account.

4. The exact annotation results corresponding to the figures could not be found in the supplemental files, which made the validation process unavailable.

5. Since tRNA fragment is also named as tRNA-derived small RNA (tsRNA for short) in several literatures (Schorn AJ, Gutbrod MJ, LeBlanc C, et al. Cell 2017; 170:61–71., Chen Q, Yan M, Cao Z, et al. Science 2016; 351:397–400.), I would suggest to stay in line with the literature, in order to maximize the visibility of the tool (and the chance of it being found through search engines).

Reviewer #2: Summary:

The manuscript by Zytnicki and Gaspin presents a newly developed small RNA annotation tool that uses a new method in order to improve small RNA annotation. Specifically it proposes a different way to handle multi-mapping sequences compared to previous methods, such as discarding multi-mapping reads, assigning ambiguous hits randomly, weighting hits (reads/n hits) or including information of expression of the hits vicinity. Their tool, mmannot, declares reads that map to different feature classes as ambiguous and provides the different assigned features to each read. As input, mmannot requires a GTF and a BAM file, while it outputs read counts for each feature, such as cds, intron, miRNA etc. The manuscript/tool is potentially a valuable addition to the field, since information on the proportion of ambiguous reads in a dataset is useful, however there are some concerns that should be addressed before publication.

Major comments:

1) Nowhere in the results do the authors discuss figure 3. Figure 3 importantly shows that the introduction of the "ambiguous" fraction does not necessarily improve the overall annotation result/overview. Here, shortstack is the tool that provides the annotation that is most similar to the "truth". As a user I would find it useful to know many reads are "ambiguous" but in the end go with shortstack anyway as it seems to be closer to the true situation. The authors should point to putative advantages of mmannot oder shortstack.

2) The authors compare mmannot to the existing tool shortstack and at some places also to MMR. However, they should compare it to other tools as well, such as the widely used featureCounts (which also takes BAM and GTF to counts features) and possibly, although this would just be a suggestion, to tools that do not use genome map files, but rather use reference sequences from several sources and databases.

Minor comments:

1) When describing the results on human and zebrafish datasets (page 4, lines 93-99 and page 5, lines 134-139) the authors provide no referral to the corresponding supplementary figures. Also the results from Fig S3 (pig) are not discussed.

2) It should be mentioned which organs/tissues the datasets are generated from.

3) Figure S2 is extremely wide. The aspect ratio of the image should be changed.

4) The order of methods for test datasets is not ideal. mmannot, BWA, mmannot re and mmannot bowtie should be grouped together.

5) The caption of figure 3 provides too little information.

6) Page 3, line 80: Remove the commas after 3, "(see Fig 2, 3, , ,)"

7) Typo in caption of Fig 2: "Our method is uses BWA..."

8) On pages 6-8 Fig 5A-F is incorrectly reffered to as Fig 4A-F

6. PLOS authors have the option to publish the peer review history of their article (what does this mean?). If published, this will include your full peer review and any attached files.

Reviewer #1: No

Reviewer #2: No

---

## [Author Response · Author response to Decision Letter 0]

16 Sep 2019

Reviewer #1: In this manuscript the authors proposed a tool to annotate small RNA based on user given annotation file. Their strategy is outputting all the annotation information to those small RNAs that

mapped to genome. Although this strategy may work in several ambiguous cases, there are several concerns before acceptance this paper.

1. The intrinsic flaw of the tool leads to the result that it can only annotate RNAs that mapped to genome. However, 10% or more RNAs cannot map to genome due to post-transcriptional splicing/modification or

other mechanism, which resulting in reduced accuracy of the tool. In this case, all the analysis in this work are biased.

  We found that at most 10% of the reads cannot be mapped on our data set, and, most of the time, the results are closer to 2-3%.

 These results have been added to Table 2.

 We cannot exclude that these missing reads would bias the results (especially if most of them are edited miRNAs, for instance), but we think that the impact on the whole classification is limited.

2. The authors mentioned piRNAs and tRNA-derived fragments in the manuscript, but didn't show them in the results, are they classified in the ambiguous category or others?

  We added the piRNA in our study, using an external database.

 They now show up in the results.

 It turns out that they do not significantly change the results, except that more reads are annotated.

 Strikingly, the number of ambiguous reads does not change significantly.

 tRNA-derived fragments, on the other hand, constitute a very small fraction of the sRNAs.

 They are thus clustered in the ``Other'' group.

3. Although the authors emphasized the advantage of their annotation strategy, the accuracy is based on the reference file the user choose. The annotation category of Ambiguous could be very high when adding

piRNA database into account.

  This is very true, and it is precisely the point of the method.

 The limitation of sRNA-Seq is that it is sometimes impossible to locus which generated a given read, without making a guess, which can be wrong.

 mmannot does not hide this and clearly states the proportion of reads which can be ambiguously mapped.

 I made your point clear in the discussion:

 ===

 A high percentage of reads produced by sRNA-Seq map to multiple loci, and this poses a major challenge to the annotation and quantification tasks.

 As a result, all the reads cannot be unambiguously annotated.

 To date, most of the tools does not report these ambiguous reads, because they remove these reads, or they weight the expression of the possible targets, using a model which can, or cannot be verified.}

 ===

 Yet, to answer your concern, we did not find that piRNAs significantly increase the number of ambiguous reads.

4. The exact annotation results corresponding to the figures could not be found in the supplemental files, which made the validation process unavailable.

  We included them in the second Supporting Information.

5. Since tRNA fragment is also named as tRNA-derived small RNA (tsRNA for short) in several literatures (Schorn AJ, Gutbrod MJ, LeBlanc C, et al. Cell 2017; 170:61–71., Chen Q, Yan M, Cao Z, et al. Science

2016; 351:397–400.), I would suggest to stay in line with the literature, in order to maximize the visibility of the tool (and the chance of it being found through search engines).

  I preferred to use the world coined by the discoverers (YS Lee, Y Shibata, A Malhotra and A Dutta, Genes and Development, 2009).

 Yet, I changed the name as you suggested.

Reviewer #2: Summary:

The manuscript by Zytnicki and Gaspin presents a newly developed small RNA annotation tool that uses a new method in order to improve small RNA annotation. Specifically it proposes a different way to handle

multi-mapping sequences compared to previous methods, such as discarding multi-mapping reads, assigning ambiguous hits randomly, weighting hits (reads/n hits) or including information of expression of the

hits vicinity. Their tool, mmannot, declares reads that map to different feature classes as ambiguous and provides the different assigned features to each read. As input, mmannot requires a GTF and a BAM

file, while it outputs read counts for each feature, such as cds, intron, miRNA etc. The manuscript/tool is potentially a valuable addition to the field, since information on the proportion of ambiguous

reads in a data set is useful, however there are some concerns that should be addressed before publication.

Major comments:

1) Nowhere in the results do the authors discuss figure 3. Figure 3 importantly shows that the introduction of the "ambiguous" fraction does not necessarily improve the overall annotation result/overview.

Here, shortstack is the tool that provides the annotation that is most similar to the "truth". As a user I would find it useful to know many reads are "ambiguous" but in the end go with shortstack anyway as

it seems to be closer to the true situation. The authors should point to putative advantages of mmannot oder shortstack.

  True.

 Actually, reads were ambiguous because they were both in the HET regions (created by the simulation tool) and the transposable elements.

 This was obviously a mistake, since the HET region is mainly transposable elements.

 We removed the transposable elements from the configuration file, and this solves the problem.

 Yet, Fig 7 now shows that reads can be mis-attributed (even if the results in general may look right).

 We made this clear in the new section ``mmannot predictions are correct''.

2) The authors compare mmannot to the existing tool shortstack and at some places also to MMR. However, they should compare it to other tools as well, such as the widely used featureCounts (which also takes

BAM and GTF to counts features) and possibly, although this would just be a suggestion, to tools that do not use genome map files, but rather use reference sequences from several sources and databases.

  We have never read a paper using featureCounts for quantifying the small RNA repertoire.

 Rather, we know it is widely used to quantify the expression of individual features (e.g. miR156).

 Moreover, featureCounts supposes that the library is either stranded or unstranded.

 As a result, it is not (directly) possible to count the reads on the reverse strand of a gene (excluding collinear reads), and the reads in either strand of the upstream region.

 Yet, we address this issue in the article, and we also implemented the other strategy you mentioned.

 The code used to generate the annotation file is given in Supporting Information.

Minor comments:

1) When describing the results on human and zebrafish data sets (page 4, lines 93-99 and page 5, lines 134-139) the authors provide no referral to the corresponding supplementary figures. Also the results

from Fig S3 (pig) are not discussed.

  We finally included these figures in the main text, and discussed them in the text.

2) It should be mentioned which organs/tissues the data sets are generated from.

  This has been added to the Table 1. We have ``transposed'' the table, for it was otherwise too wide.

3) Figure S2 is extremely wide. The aspect ratio of the image should be changed.

  We now provide only the five first samples. The results are very similar, and the other samples do not provide much more information.

4) The order of methods for test data sets is not ideal. mmannot, BWA, mmannot re and mmannot bowtie should be grouped together.

  Done.

5) The caption of figure 3 provides too little information.

  We have added this in the caption.

 The results of all but the last bars is explained in Fig 2.

 The ``truth'' bar gives the fraction of each sRNA class generated by our method.

6) Page 3, line 80: Remove the commas after 3, "(see Fig 2, 3, , ,)"

  For some unknown reason, LaTeX does not seem to cross-reference right.

 We hard-coded the missing references.

7) Typo in caption of Fig 2: "Our method is uses BWA..."

  Thanks, corrected!

8) On pages 6-8 Fig 5A-F is incorrectly reffered to as Fig 4A-F

  This was probably a problem issued by the PLOS LaTeX compiler.

 On pages 6-8, there are no mention of Fig 4, only Fig 5.

 We will try to be careful during the proof production.

---

## [Decision Letter · Decision Letter 1]

1 Oct 2019

PONE-D-19-16932R1

mmannot: How to improve small–RNA annotation?

PLOS ONE

Dear Dr Zytnicki,

Thank you for submitting your manuscript to PLOS ONE. After careful consideration, we feel that it has merit but does not fully meet PLOS ONE’s publication criteria as it currently stands. Therefore, we invite you to submit a revised version of the manuscript that addresses the points raised during the review process.

One reviewer still have major concerns about this revised version. Please carefully revise this manuscript again following the reviewers' suggestion.

We would appreciate receiving your revised manuscript by Nov 15 2019 11:59PM. To enhance the reproducibility of your results, we recommend that if applicable you deposit your laboratory protocols in protocols.io, where a protocol can be assigned its own identifier (DOI) such that it can be cited independently in the future. For instructions see: http://journals.plos.org/plosone/s/submission-guidelines#loc-laboratory-protocols

We look forward to receiving your revised manuscript.

Kind regards,

Francesca Rizzo, PhD

Academic Editor

PLOS ONE

Reviewers' comments:

Reviewer's Responses to Questions

**Comments to the Author**

1. If the authors have adequately addressed your comments raised in a previous round of review and you feel that this manuscript is now acceptable for publication, you may indicate that here to bypass the “Comments to the Author” section, enter your conflict of interest statement in the “Confidential to Editor” section, and submit your "Accept" recommendation.

Reviewer #1: (No Response)

Reviewer #2: (No Response)

2. Is the manuscript technically sound, and do the data support the conclusions?

Reviewer #1: No

Reviewer #2: Partly

3. Has the statistical analysis been performed appropriately and rigorously? 

Reviewer #1: No

Reviewer #2: N/A

4. Have the authors made all data underlying the findings in their manuscript fully available?

Reviewer #1: Yes

Reviewer #2: Yes

5. Is the manuscript presented in an intelligible fashion and written in standard English?

Reviewer #1: No

Reviewer #2: Yes

6. Review Comments to the Author

Reviewer #1: The revised manuscript by Zytnicki and Gaspin is improved indeed. However, my concerns still exist before consider to recommend this work to get published.

Major concerns:

1. The authors “think” the unmapped rate is as low as 2-3% and the impact is limited based on their selected datasets. However, this conclusion is not true for some, if not for most datasets (e.g. datasets mentioned in RNA Biology, 11(11), 1375–1385. doi:10.1080/15476286.2014.996465, the unmapped rate is as high as 27%-48%). Since they didn’t solve this problem in the revised manuscript, the intrinsic flaw of the tool is still existing.

2. The authors may not claim their annotation method is a correct one by only using a simulated data set, as well as the annotation of a sequence is hard to define correctly without determining their function in a biological way. In this case, they may change the subtitle ''mmannot predictions are correct'' to a more appropriate one, maybe ''mmannot predictions are more comprehensive”.

3. The authors claimed they did not find that piRNAs significantly increase the number of ambiguous reads, while it might be true in their selected datasets, I am highly suspect if the authors chose the appropriate piRNA database or have done the appropriate analysis in their work. According to this published work (Commun Biol. 2018 Jan 22;1:2. doi: 10.1038/s42003-017-0001-7) and my knowledge, the overlapping annotations between piRNA and other RNA species are not negligible.

Minor concerns:

1. Fig 1 is missing in the revised manuscript.

2. Please clarify which piRNA database the authors used in this work.

Reviewer #2: Most concern have been adequately addressed by the authors. However, some minor issues are still present or have newly arisen within the revision of the manuscript.

Page 1, line 9: Typo, "nuclear" not "nuclar"

Page 1, line 9: Typo, lacking "."

Page 3, lines 99/100: "For all data sets (see Fig 2 to 5), the miRNA and the piRNA classes are generally the 99 most represented whatever the used strategy." I am sure the authors don't want to imply this, but Plants (as A. thaliana (Fig 2)) do not have piRNAs. The sentence gives this false impression, however.

Page 5, lines 136/137: "The “random” strategy randomly places reads. As seen in Fig 7, this method yields a 136 high number of false positives." This seems to be the case in the old version of the figure, but not in the current version. #FP is very low.

Page 6, line 173: Typo, "piRNA annotation" not "piRNA annotion"

Page 6, line 177: "As seen on Fig 7, only mmannot and the “sequence” strategy give no false positives." This is confusing and does not seem to overlap with the content of the figure. Do the authors mean false negatives?

Page 6, line 208: Typo, "do" not "does"

Page 10, Table 1: It's an improvement that the authors include tissues of the data sets. However: 1. "several" for D. rerio is no gain in information. I would also suggest to include the tissues within the figures for each individual data set.

Figure 3: It is unlikely that human brain tissues, such as amygdala, would contain any piRNAs, but 30/50%-90% seems highly doubtful. The database used here seems flawed. Often such databases wrongly include miRNAs (as the authors mention) but also tRNA-fragments, among others. This worsens mmannot's annotation. Figure 5 - Sus scrofa testis - however, looks realistic concerning piRNA content. Presumably this piRNA database can only be reliably used on germline tissues.

7. PLOS authors have the option to publish the peer review history of their article (what does this mean?). If published, this will include your full peer review and any attached files.

Reviewer #1: No

Reviewer #2: No

---

## [Author Response · Author response to Decision Letter 1]

27 Nov 2019

Dear reviewers, we would like to thank you for providing us crucial feedback in order to improve the quality of the article.

We tried to address all the questions they had, especially about the read mapping rate, at the end of this file.

Best regads,

Christine Gaspin and Matthias Zytnicki.

===

Reviewer #1: The revised manuscript by Zytnicki and Gaspin is improved indeed. However, my concerns still exist before consider to recommend this work to get published.

Major concerns:

1. The authors “think” the unmapped rate is as low as 2-3\\% and the impact is limited based on their selected datasets. However, this conclusion is not true for some, if not for most datasets (e.g. datasets mentioned in RNA Biology, 11(11), 1375–1385. doi:10.1080/15476286.2014.996465, the unmapped rate is as high as 27%-48%). Since they didn’t solve this problem in the revised manuscript, the intrinsic flaw of the tool is still existing.

  We downloaded five data sets used in the paper (actually, all the data sets where the mapping rate was provided in detail): SRR039615, SRR191522, SRR191529, SRR191552, and SRR372670.

 We used the following commands for each fastq file:

 - fastx_clipper -a <adapter> -l 15 -i file.fastq > file_trimmed1.fastq

 - fastq_quality_trimmer -l 15 -t 20 < file_trimmed1.fastq > file_trimmed2.fastq

 - fastq_quality_filter -q 20 -p 80 < file_trimmed2.fastq > file_trimmed3.fastq

 - fastq2fasta < file_trimmed3.fastq > file_trimmed3.fasta

 - dust file_trimmed3.fasta > file_trimmed4.fasta

 - fasta2fastq < file_trimmed4.fasta > file_trimmed4.fastq

 - bowtie -S -a --best --strata -p 6 database file_trimmed4.fastq > file.sam

 (The adapters are TCGTATGCCGTCTTCTGCTTG TTGGTTCGTATGCCGTC TCCACTCGTATGCCGTCTTT TCATCTCGTATGCCGTCTTC ATCTCGTATGCCGTCTTC)

 We had the following results.

 # reads # trimmed % not mapping

 SRR039615 9,288,038 7,415,686 1.32%

 SRR191522 71,025 57,895 2.50%

 SRR191529 11,730 7,356 4.69%

 SRR191552 385,594 287,964 5.56%

 SRR372670 48,415,574 2,8822,982 17.17%

 It is true that the last sample did not map well, however, we could not confirm an unmapping rate of 27%-48%.

 It is also likely that the trimming we used might be more drastic that the one used in the paper.

 What we would like to stress here, is that a mapping rate of <5% is now not uncommon, provided that the reads have been efficiently trimmed.

2. The authors may not claim their annotation method is a correct one by only using a simulated data set, as well as the annotation of a sequence is hard to define correctly without determining their function in a biological way. In this case, they may change the subtitle ''mmannot predictions are correct'' to a more appropriate one, maybe ''mmannot predictions are more comprehensive”.

  True, we should not over-sell our method. We made the suggested modification. We also changed "unbiased" to "less biased."

3. The authors claimed they did not find that piRNAs significantly increase the number of ambiguous reads, while it might be true in their selected datasets, I am highly suspect if the authors chose the appropriate piRNA database or have done the appropriate analysis in their work. According to this published work (Commun Biol. 2018 Jan 22;1:2. doi: 10.1038/s42003-017-0001-7) and my knowledge, the overlapping annotations between piRNA and other RNA species are not negligible.

 This is very true, and we made this clear in the article (already in the previous revision):

 [...] we compared the miRNA and the piRNA annotations.

 It turns out that 1178 out of the 3934 annotated miRNAs overlap a piRNA annotation.

 The most likely reason is that a great fraction of the piRNAs are probably mis-annotated

 In fact, we simply used the annotation from published database "piRBase".

 We felt that discovering piRNAs transcripts was outside the scope of the analysis.

Minor concerns:

1. Fig 1 is missing in the revised manuscript.

  Probably a problem during the upload. We corrected this.

2. Please clarify which piRNA database the authors used in this work.

  We used the annotation from piRBase. We added the following in the manuscript:

 For D. rerio, H. sapiens, and S. scrofa, we downloaded a piRNA annotation from piRBase, and we added this annotation to the existing annotations (which usually do not include piRNAs).

Reviewer #2: Most concern have been adequately addressed by the authors. However, some minor issues are still present or have newly arisen within the revision of the manuscript.

Page 1, line 9: Typo, "nuclear" not "nuclar"

  Done.

Page 1, line 9: Typo, lacking "."

  Done.

Page 3, lines 99/100: "For all data sets (see Fig 2 to 5), the miRNA and the piRNA classes are generally the 99 most represented whatever the used strategy." I am sure the authors don't want to imply this, but Plants (as A. thaliana (Fig 2)) do not have piRNAs. The sentence gives this false impression, however.

  Very well, we removed this sentence.

Page 5, lines 136/137: "The “random” strategy randomly places reads. As seen in Fig 7, this method yields a 136 high number of false positives." This seems to be the case in the old version of the figure, but not in the current version. #FP is very low.

  True, we meant "false negatives". This has been corrected.

Page 6, line 173: Typo, "piRNA annotation" not "piRNA annotion"

  Done.

Page 6, line 177: "As seen on Fig 7, only mmannot and the “sequence” strategy give no false positives." This is confusing and does not seem to overlap with the content of the figure. Do the authors mean false negatives?

 True, we also meant "false negatives". This has also been corrected.

Page 6, line 208: Typo, "do" not "does"

  Done.

Page 10, Table 1: It's an improvement that the authors include tissues of the data sets. However: 1. "several" for D. rerio is no gain in information. I would also suggest to include the tissues within the figures for each individual data set.

  We forgot to mention that the samples that we used here were extracted from brain tissue. We corrected this on the manuscript.

Figure 3: It is unlikely that human brain tissues, such as amygdala, would contain any piRNAs, but 30/50%-90% seems highly doubtful. The database used here seems flawed. Often such databases wrongly include miRNAs (as the authors mention) but also tRNA-fragments, among others. This worsens mmannot's annotation. Figure 5 - Sus scrofa testis - however, looks realistic concerning piRNA content. Presumably this piRNA database can only be reliably used on germline tissues.

  This is very true. We mentioned this in the "Results" section

 Using the expected configuration file, the number of miRNAs is strictly greater than the number of piRNA.

 However, using the reverse configuration file, the number of piRNAs is much greater.

 To understand why, we compared the miRNA and the piRNA annotations.

 It turns out that 1178 out of the 3934 annotated miRNAs overlap a piRNA \\hl{annotation}.

 The most likely reason is that a great fraction of the piRNAs are probably mis-annotated, and using a meaningful configuration file is sometimes crucial for the analysis of small RNAs.

 To make it even more clear, we also added a new paragraph in the "Discussion":

 By nature, mmannot heavily rely on the accuracy of the available annotation.

 We saw, on the human data set, that the piRNA annotation might be flawed, and this dramatically alters the read annotation produced by our method.

 However, a simple, common-sense, ordering of the annotation types (favor ``miRNA'' when a read overlaps a miRNA annotation and a piRNA annotation) produces the expected results.

 Alternatively, discarding unexpected annotations (\\textit{e.g.} piRNAs in brain) might also be a sensible method.

---

## [Decision Letter · Decision Letter 2]

18 Dec 2019

PONE-D-19-16932R2

mmannot: How to improve small–RNA annotation?

PLOS ONE

Dear Dr Zytnicki,

Thank you for submitting your manuscript to PLOS ONE. After careful consideration, we feel that it has merit but does not fully meet PLOS ONE’s publication criteria as it currently stands. Therefore, we invite you to submit a revised version of the manuscript that addresses the points raised during the review process.

We would appreciate receiving your revised manuscript by Feb 01 2020 11:59PM. To enhance the reproducibility of your results, we recommend that if applicable you deposit your laboratory protocols in protocols.io, where a protocol can be assigned its own identifier (DOI) such that it can be cited independently in the future. For instructions see: http://journals.plos.org/plosone/s/submission-guidelines#loc-laboratory-protocols

We look forward to receiving your revised manuscript.

Kind regards,

Francesca Rizzo, PhD

Academic Editor

PLOS ONE

Reviewers' comments:

Reviewer's Responses to Questions

**Comments to the Author**

1. If the authors have adequately addressed your comments raised in a previous round of review and you feel that this manuscript is now acceptable for publication, you may indicate that here to bypass the “Comments to the Author” section, enter your conflict of interest statement in the “Confidential to Editor” section, and submit your "Accept" recommendation.

Reviewer #1: All comments have been addressed

Reviewer #2: All comments have been addressed

2. Is the manuscript technically sound, and do the data support the conclusions?

Reviewer #1: Yes

Reviewer #2: Yes

3. Has the statistical analysis been performed appropriately and rigorously? 

Reviewer #1: Yes

Reviewer #2: Yes

4. Have the authors made all data underlying the findings in their manuscript fully available?

Reviewer #1: Yes

Reviewer #2: Yes

5. Is the manuscript presented in an intelligible fashion and written in standard English?

Reviewer #1: Yes

Reviewer #2: Yes

6. Review Comments to the Author

Reviewer #1: Based on the parameters the authors provided in the response, I understand that the mapping rate issue based on the reference database they chose to map. In this way, the actual annotation rate might be provided to clarify the advantage of their software among others.

Reviewer #2: (No Response)

7. PLOS authors have the option to publish the peer review history of their article (what does this mean?). If published, this will include your full peer review and any attached files.

Reviewer #1: No

Reviewer #2: No

---

## [Author Response · Author response to Decision Letter 2]

30 Jan 2020

Dear reviewer,

We would like to thank you for providing us crucial feedback in order to improve the quality of the article.

As suggested, we computed the actual annotation rates, which are the number of reads used for the annotation, divided by the number of reads sequenced.

We added a figure with all the annotation rates, and it confirms that almost all the reads can be used for annotating the genome, as long as the sequencing quality is good, and the annotation exhaustive.

Best regards,

Christine Gaspin and Matthias Zytnicki.

---

## [Decision Letter · Decision Letter 3]

19 Feb 2020

PONE-D-19-16932R3

mmannot: How to improve small–RNA annotation?

PLOS ONE

Dear Dr Zytnicki,

Thank you for submitting your manuscript to PLOS ONE. After careful consideration, we feel that it has merit but does not fully meet PLOS ONE’s publication criteria as it currently stands. Therefore, we invite you to submit a revised version of the manuscript that addresses the points raised during the review process.

We would appreciate receiving your revised manuscript by Apr 04 2020 11:59PM. To enhance the reproducibility of your results, we recommend that if applicable you deposit your laboratory protocols in protocols.io, where a protocol can be assigned its own identifier (DOI) such that it can be cited independently in the future. For instructions see: http://journals.plos.org/plosone/s/submission-guidelines#loc-laboratory-protocols

We look forward to receiving your revised manuscript.

Kind regards,

Francesca Rizzo, PhD

Academic Editor

PLOS ONE

Reviewers' comments:

Reviewer's Responses to Questions

**Comments to the Author**

1. If the authors have adequately addressed your comments raised in a previous round of review and you feel that this manuscript is now acceptable for publication, you may indicate that here to bypass the “Comments to the Author” section, enter your conflict of interest statement in the “Confidential to Editor” section, and submit your "Accept" recommendation.

Reviewer #1: All comments have been addressed

2. Is the manuscript technically sound, and do the data support the conclusions?

Reviewer #1: Partly

3. Has the statistical analysis been performed appropriately and rigorously? 

Reviewer #1: No

4. Have the authors made all data underlying the findings in their manuscript fully available?

Reviewer #1: Yes

5. Is the manuscript presented in an intelligible fashion and written in standard English?

Reviewer #1: Yes

6. Review Comments to the Author

Reviewer #1: Most of my concerns are addressed in the latest revised manuscript. The new figure 7 confirmed my points that the annotation rate varies in different species. A statistical analysis might be performed among different software in this figure.

7. PLOS authors have the option to publish the peer review history of their article (what does this mean?). If published, this will include your full peer review and any attached files.

Reviewer #1: No

---

## [Author Response · Author response to Decision Letter 3]

25 Mar 2020

Dear Reviewer,

First, we would like to thank you for providing us crucial feedback in order to improve the quality of the article.

We counted the number of reads used by our method and other methods, and we tested (using a one-sided Wilcoxon signed-rank test) if we used significantly more reads.

The p-value are always not greater than 2e-4.

These results are summarized in a new Table, which we added in the manuscript.

We hope that you will find this revision acceptable.

Best regads,

Christine Gaspin and Matthias Zytnicki.

---

## [Editor Report · Decision Letter 4]

31 Mar 2020

mmannot: How to improve small–RNA annotation?

PONE-D-19-16932R4

Dear Dr. Zytnicki,

We are pleased to inform you that your manuscript has been judged scientifically suitable for publication and will be formally accepted for publication once it complies with all outstanding technical requirements.

With kind regards,

Francesca Rizzo, PhD

Academic Editor

PLOS ONE

---

## [Editor Report · Acceptance letter]

3 Apr 2020

PONE-D-19-16932R4 

mmannot: How to improve small–RNA annotation? 

Dear Dr. Matthias:

I am pleased to inform you that your manuscript has been deemed suitable for publication in PLOS ONE. Congratulations! Your manuscript is now with our production department. 

With kind regards,

on behalf of

Dr. Francesca Rizzo 

Academic Editor

PLOS ONE